# Virtual Autonomous Driving with Reinforcement Learning

**Ziwen Lu**
Department of Information Engineering
The Chinese University of Hong Kong
Shatin, Hong Kong
1155155161@link.cuhk.edu.hk

**Ziqi Wang**
Department of Information Engineering
The Chinese University of Hong Kong
Shatin, Hong Kong
1155155096@link.cuhk.edu.hk

## Abstract

Autonomous driving is becoming the trend for future transportation. One of its most significant challenges is to recognize traffic signs and obey traffic rules specified by the signs. In this paper, the particular topic of optimal vehicle speed control whenever a vehicle reaches a speed limit sign is studied. This research is conducted in a longitudinal environment, which only has one dimension, the straight traffic lane, along which a vehicle will drive through. There will be multiple traffic signs in the traffic lane, which the vehicle can recognize 150 meters in advance. Three factors are taken into account during control optimization, that is, energy consumption of the vehicle, jerk (change of acceleration), and most importantly, speed of the vehicle. Methods of Q learning with the deep neural network are implemented in the research for speed limit control. More concisely, three deep learning methods are implemented, which are DDPG, TD3 and SAC. The last of the three is the default method for this environment and thus will serve as the baseline. Experiment results show that TD3 and SAC algorithms both achieved comparably high performance within five-hour training span. Specifically, TD3 gained larger policy improvement per epoch of training while SAC achieved faster training efficiency. On the other hand, DDPG achieved worse performance than the other two algorithms due to its instability during training and slow training efficiency. The presentation link is "zoom video".

## 1 Introduction

The main objective of this study is to achieve automatic speed limit control with deep learning methods. The experiments are conducted in a longitudinal environment called "the LongiControl Environment". In this environment, an electric vehicle will drive along a single straight traffic lane, on which only speed limit traffic signs exist, as shown in figure 1 below. [1][4]Factors of traffic conditions will not be studied in the scope of this paper. The expected result after training is that the agent, in this case, the vehicle, will drive as fast and smooth as possible within the speed limit, while also as energy efficient as possible. Hence, to reflect the stated goal, rewards from the environment consist of four parts, the speed difference between the current speed and speed limit, the energy consumption of the vehicle, the jerk (change of acceleration), and lastly, the shock (penalty for speeding). It is worth noting that the agent in our experiment is modelled after a real-world

vehicle, that is, the performance and energy consumption of the vehicle all comes from experimental results of a 2014 BMW i3. With regard to the training algorithm, three deep Q learning methods are implemented, including DDPG, TD3 and SAC. These three algorithms are similar in that they both need a critical network to guild the optimization direction and an actor network that directly optimize the policy. Difference between them will be explained in detail below. The SAC algorithm is the default set by the LongiControl author so it will be used as our baseline method. After five hours of training, SAC achieved a reward of -112.4 with 1000 epochs of training, TD3 achieved a reward of -134.2 with 595 epochs of training, and DDPG achieved a reward of -267.8 with 155 epochs of training. The rewards of the three algorithms showed that TD3 reached a good performance comparable to the SAC method, while DDPG did not perform as well as the other two algorithms. Regarding training efficiency, SAC ran more epochs of training than TD3 within the same period of time, while TD3 gained larger improvements per epoch of training. Contrarily, DDPG had the slowest training efficiency and the worst performance among the three, mainly due to training instability.[3]

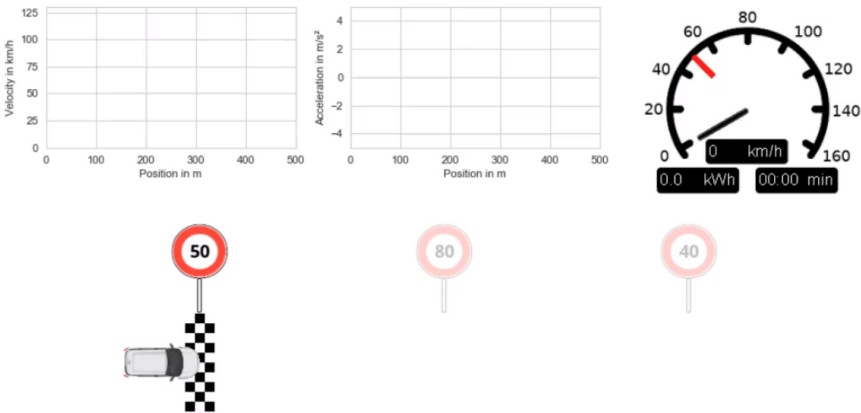

Figure 1: The "Longicontrol Environment". In this environment, an electric vehicle will drive along a single straight traffic lane, on which only speed limit traffic signs exist.

## 2 Problem Formulation

### 2.1 Environment setting

In order to evaluate the agent's performance, the environment will record the agent's speed, previous and current acceleration, current power consumption and current speed limit. It will then give rewards as a combination of the energy consumption, difference between current speed and speed limit, jerk and shock. The detailed rewards are calculated as follow:

$$r_t = -w_1(P_t/p_{max})dt - w_2((v_t - v_{limit})/v_{limit}) - w_3((a_t - a_{t-1})/jerk) - w_4 shock \quad (1)$$

Where $p_t$ is the current power and $p_{max}$ is the max power of the vehicle, $dt$ is the time difference, $v_t$ and $v_{limit}$ are current speed and speed limit, $a_t$ and $a_{t-1}$ are current acceleration and previous acceleration, jerk is the maximum allowed jerk set by the user, shock is 0 when not speeding and 1 when speeding, and the four w are weights of each component, default as 1, 0.5, 1, 1. The goal of the agent is to maximize the total rewards, which is the sum of rewards over the entire trajectory.

### 2.2 Agent setting

At each time step, the agent will be able to observe his state from the environment, which consists of its current speed, previous acceleration, current speed limit, future speed limit, and the distance from the next speed limit signs. It will then give action value between [-1,1], where the negative sign denotes deceleration and the positive sign denotes acceleration, and the magnitude corresponds to magnitude of acceleration. The action value will then be passed into the real-world vehicle model

and output the actual acceleration. With the acceleration, the speed and distance traveled can be calculated as follow:

$$v_t = a_t dt + v_{t-1} \tag{2}$$

$$x_t = 1/2 a_t (dt)^2 + v_{t-1} dt + x_{t-1}(3) \tag{3}$$

Where $v_t$ and $v_{t-1}$ are the current speed and the previous speed, $dt$ is the time difference, $x_t$ and $x_{t-1}$ are the current and previous travelled distance. When the agent travels 1000 meters, the current trajectory is considered done and the next epoch of training will start.

## 2.3 Methodology

A brief introduction about the three deep Q learning methods will be given for better understanding of our research topic. All three deep Q learning algorithms use the actor critic strategy for a more stable policy update. For both DDPG and TD3 algorithms, the objective is to minimize value prediction error of the critic and maximize Q values from the actor, illustrated by the two equations below:

critic network:

$$Min(r + Q_{target}(s, \mu) - Q(s, \mu)) \tag{4}$$

actor network:

$$Max(Q(s, \mu)) \tag{5}$$

Where $\mu$ is the action of the agent decided by the current policy, $r$ is the reward for the current timestamp and $Q_{target}$ is target Q value generated by the target network, which is the delayed critic network. The major difference between the two algorithms is that TD3 uses two critic networks for updating target Q value and takes the smaller of the two to calculate TD error, while DDPG only uses one critic network. The actor network will directly output the action value $Q(s, \mu)$. The SAC method is similar to the TD3 method, except that it takes into account entropy when optimizing critic network and actor network. Equations for SAC are given as follow:

critic network:

$$Min(r + Q_{target}(s, \mu) - \alpha log(\pi(s, \mu)) - Q(s, \mu)) \tag{6}$$

actor network:

$$Max(Q(s, \mu)) - \alpha log(\pi(s, \mu)) \tag{7}$$

Where the $log(\pi(s, \mu))$ is the entropy and $\alpha$ is the trade-off coefficient set by the user.

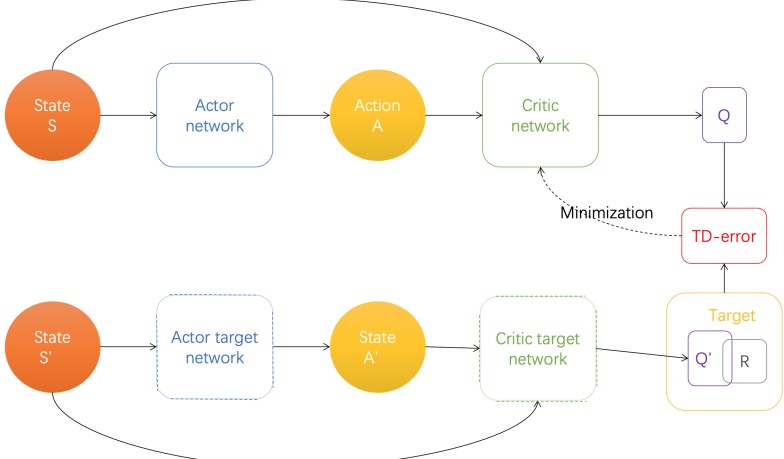

Figure 2: The network architecture of DDPG: The Actor needs to find A to maximize the output Q. For critic, DDPG learns one Q-functions, and uses this Q-values to form the target in the Bellman error loss functions. Then, this target will update the Critic network.

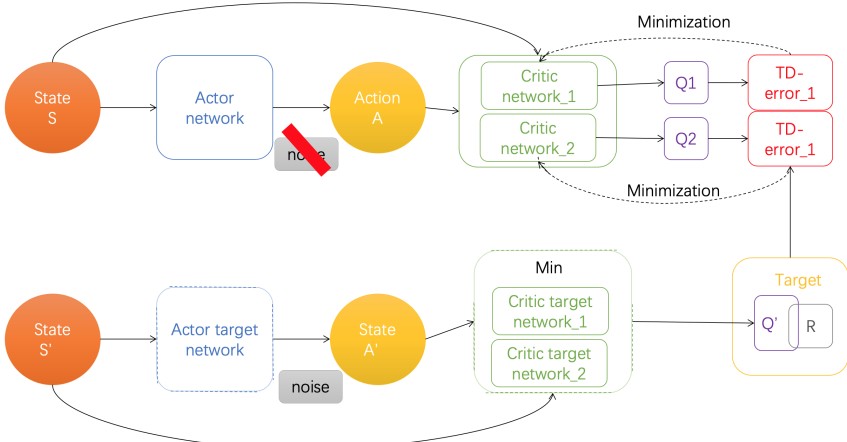

Figure 3: The network architecture of TD3 and SAC: The Actor needs to find A to maximize the output $Q$. TD3 adds noise to the target action, to make it harder for the policy to exploit $Q-$function errors by smoothing out $Q$ along changes in action. For critic, TD3 learns two $Q-$functions instead of one, and uses the smaller of the two $Q-$values to form the target in the Bellman error loss functions. Then, this target will update two networks Critic network 1 and Critic network 2.The SAC algorithm uses two $Q-$networks structure of TD3.

## 3   Experiment Result

All three algorithms were trained for five hours in the "LongiControl Environment".[2] After training, SAC increased the reward from -1741.58 to -162.29, TD3 escalated the reward from -2140.00 to -167.42 and DDPG improved the reward from -2094.25 to -297.61. The improvement of the reward during the training process is shown below in figure 4.

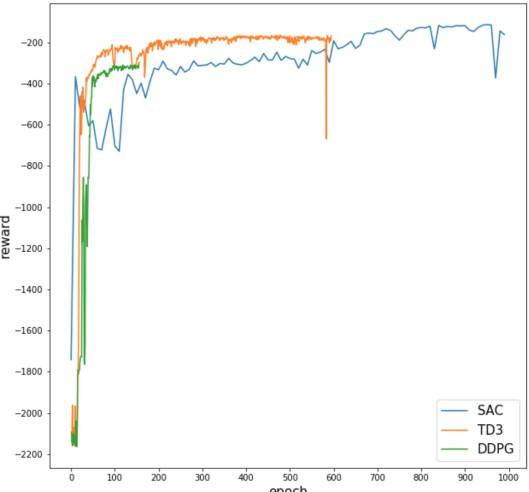

Figure 4: Relation between reward and number of training epochs for SAC, TD3 and DDPG algorithms. The reward increased dramatically during the beginning of the training process. After a period of fluctuation, the increase became slow but steady. Differences between the number of epochs were caused by different training efficiency among the three algorithms.

From figure 4, one can infer that the TD3 algorithm achieved well performance comparable to the benchmark SAC algorithm, despite almost fifty percent less training epochs. To be specific, SAC had a better training efficiency while TD3 had larger policy improvement per epoch of training. In

contrast, DDPG achieved the worst rewards due to its slow training efficiency and unstable training behavior.

In order to visualize the performance of the agent, one epoch of the test was run for the three algorithms and the results were rendered in the following three figures. For each plot, the subplot at the top left showed the speed of the vehicle as a function of its position, and the red dash line indicates the speed limit. Subplot in the middle recorded the acceleration as a function of position. Support on the right gave information about vehicle current speed, power consumption and operation time.

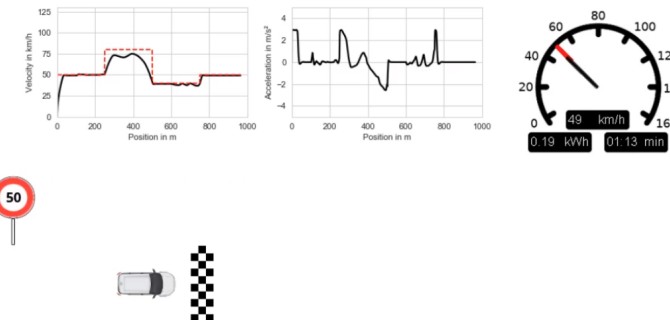

Figure 5: Rendered result for SAC

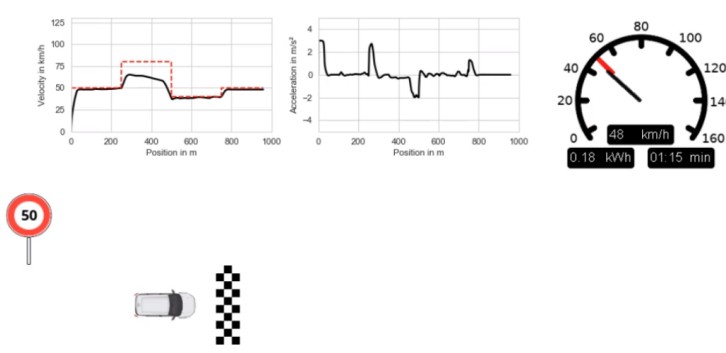

Figure 6: Rendered result for TD3

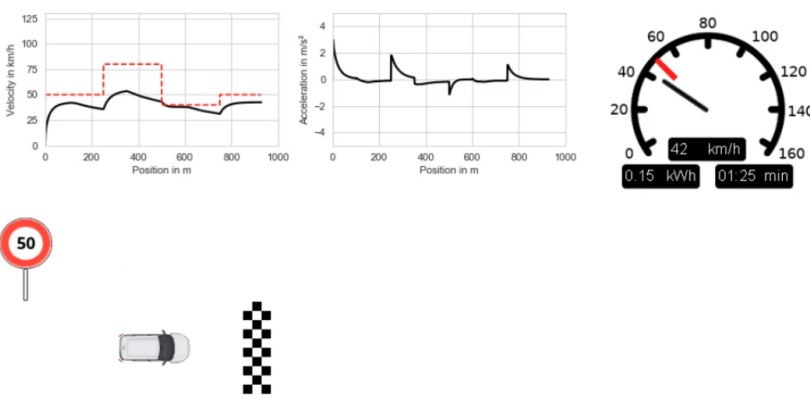

Figure 7: Rendered result for DDPG

The figures above testifies the effectiveness of the training of both SAC and TD3. Compared to SAC, TD3 yielded a smaller jerk when the vehicle reached a new speed limit sign, while SAC produced a

larger jerk and a speed closer to the speed limit. Both algorithms strictly obey speed limits to avoid any shock. In contrast, DDPG yielded an inferior control result in that the vehicle speed deviated far from the speed limit and it violated the speed limit when reaching a new speed limit sign.

## 4   Conclusion

Our project explores the autonomous speed control problem with three deep Q learning algorithms, which are SAC, TD3 and DDPG. Experiments were conducted in the "Longicontrol Environment" in which a vehicle will drive through a single straight lane where speed limit signs exist. Our experiment results show that both the SAC and TD3 algorithms achieved good control performance, with TD3 yielding a smoother ride (less jerk) and SAC more precise control (speed closer to the speed limit). On the other hand, DDPG gained an inferior performance due to its low training efficiency and training instability.

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
