# OpenReview forum: "Virtual Autonomous Driving with Reinforcement Learning"
_CUHK.edu.hk/2021/Course/IERG5350_

### Official Review · AnonReviewer3 · 2020-12-15
**Valuable investigation, but some real cases should be included in the experiment......**

**Rating:** 7
**Confidence:** 3

**Review:**

I am not very familiar with the relevant literature, but I tried to make my review honestly based on the report and what I know.

Significance: This paper is to investigate Virtual Autonomous Driving with Reinforcement Learning. And the conclusion is TD3 and SAC can achieve very high performance in the optimal vehicle speed control. This will be significance as controlling speed of an autonomous car is an important for the traffic flow and safely. (As too fast or too slow may cause traffic accident.)

Novelty: It is proposing one of the solutions to due with the problem, but from the reference you may just doing from a given environment by others. However it is still have the different training process, it is still a fair innovation.

Technical quality: Good. Clear defined formula and methodology.

Clarity: It is clear to read, well-organized and able to follow your logic. Especially the figures that provided by you guys.

My some comments or suggestions:

(1)	Where is your source code?

(2)	It would be better if you can discuss more on the result. (eg. Why DDPG have a instability during training and slow training efficiency? Due to environment? Or …… This may help you to find the best alg. to due with this problem.)

(3)	Although it is fine for your paper, but I hope you can have some more innovations. May be this is too cruel to you: First in real case on road construction, for example in Hong Kong, there are road sign to warn you ready to reduce the speed. (Like Warning sign in https://www.td.gov.hk/en/road_safety/road_users_code/index/chapter_5_for_all_drivers/roads_with_faster_traffic_/index.html) As the suddenly reduce speed have the danger to have collision, this also show in your result graph. Is that your agent can detect this and start to reduce the speed, although you can see the road sign in a distance of 150 meters? Second, it may be good to have padding case, like some cars are in the front to test your agent about the performance to see whether the agent will be affected by the traffic flow.

Actual rating I did is 6.5, but no 6.5......

---

### Official Review · AnonReviewer2 · 2020-12-16
**This paper studies several RL algorithms in the speed limit control problem of Autonomous Driving**

**Rating:** 6
**Confidence:** 3

**Review:**

General:
This paper studies the speed limit control problem of the Autonomous Driving field. Several methods including DDPG, TD3 and SAC were implemented in the virtual environment.

Comments:
1.Technical quality: Clear defined formula and methodology.
2.Clarity: The paper is well-organized and easy to read. Figures you provided help us better understand your explanation and experiment.
3. It seems that you set a fixed training time and explore three algorithms's training efficiency and results. Hope you can do more experiments. For example, don't fix training time, just see how these three methods can performance with enough training steps.

---

### Official Review · AnonReviewer1 · 2020-12-20
**I recommend the authors to do more experiments and analysis to get a better rating score.**

**Rating:** 5
**Confidence:** 4

**Review:**

# Summary
The main work of this paper is to apply three RL algorithms, SAC, DDPG, TD3 into a Virtual Autonomous Driving Environment called LongiControl. The experiments show that the SAC outperforms the other two algorithms, which explains the reason why the author of LongiControl chooses SAC as Agent.

# Novelty
The novelty of this paper is limited, as they didn't make any changes to the algorithm or environment which can improve the original work proposed by LongiControl paper.

# Significance
Virtual Autonomous Driving is an important and hot topic, which has the potential to make a change in our life. Therefore the topic the author chose is significant.

# Strong Points
1) The author tried more algorithms than the author of LongiControl.
2) The author has a good understanding of RL algorithms.

# Week Points
1) The novelty and significance of this paper are limited. The author seems just used the default parameters of the three algorithms and didn't try to improve them by tuning or modification (at least they didn't show me in the paper).
2) The author didn't clarify the intuition of the definition of reward and the weights choosing, which is very important to reinforcement learning.
3) There is no code provided in paper or video, which makes it harder to judge the workload of this paper.

# Revision Advice
1) Clarify your environment design or show me your code.
2) Clarify your reason for defining the Reward function like that. (I'm confused why the acceleration is taken into consideration.)
3) I think you can do more analysis on your experiment results, e.g. why DDPG has bad performance in both efficiency and accuracy. This can show that you really understand this topic or even made some contributions to it.
4) Maybe you can get better performance with TD3 if you can tuning hyper-parameters of it.